# Analysis of the Possibility of Using Cenospheres in the Production of Cement Mortars for Use in an Elevated Temperature Environment

**DOI:** 10.3390/s22197518

**Published:** 2022-10-04

**Authors:** Gabriela Rutkowska, Paweł Ogrodnik, Mariusz Żółtowski, Aleksandra Powęzka, Karolina Kaszewska

**Affiliations:** 1Institute of Civil Engineering, Warsaw University of Life Sciences, 166 Nowoursynowska Street, 02-787 Warsaw, Poland; 2Faculty of Safety Engineering and Civil Protection, The Main School of Fire Service, 52/54 Słowackiego Street, 01-629 Warsaw, Poland

**Keywords:** cement mortars, cenospheres, strength, temperature

## Abstract

The topic of research included in this article was the evaluation of the influence of cenospheres on selected parameters of mortar cement. Samples were designed as CEM I 42.5 R Portland cement with the application of different additive amounts. In the experimental work, the consistency, compressive strength, and bending strength were tested after 28 and 56 days of maturation, and after heating temperatures of 20, 300, 500, and 700 °C. The compressive strength was tested on half beams (40 × 40 × 160 mm). Using the obtained results, the properties of the mortars were compared. The research confirmed the possibility of producing cenosphere-modified cement mortars. Cenospheres used in the preparation of cement mortar negatively affected the bending and compressive strength with increasing temperature (20, 300, 500, 700 °C) and increasing content of this additive (10, 20, 30%).

## 1. Introduction

Coal is currently one of the world’s basic energy resources, and will remain so for the next few years [1]. In Poland, about 90% of electricity comes from coal. Its combustion in industrial processes is a major source of CO_2_, SO_2_, NOx and dust emissions. To maintain sustainable development, and thus a clean environment, increasingly stringent requirements on pollutant emissions standards have been imposed by the EU [2]. Emerging clean coal technology (CTW) processes are associated with the production of waste that requires management or disposal. Due to the dynamic development of CTW processes and technological progress, the waste in question is treated as a valuable product. Because furnaces operate at high temperatures (1000–1800 °C), the organic matter of the coal combusted is decomposed into molten slag of morphologically variable size and shape [3]. This variability leads to the formation of various types of particles in coal fly ash (CFA), such as plerospheres, ferrospheres, and cenospheres. Figure 1 illustrates cenospheres. A necessary condition for the formation of aluminosilicate microspheres is the presence of gases trapped inside the molten ash droplet [4,5].

The properties of aluminosilicate microspheres depend on the type and quality of coal, and the temperature and time in the combustion and cooling zones. Plerospheres are large microspheres with encapsulated particles filled with fine clusters of spherical particles along with minerals and gases [5], while ferrospheres are spherical particles rich in iron [6]. Cenospheres (CS), similar to fly ash, are mainly aluminosilicate spheres filled with gases from the coal combustion process: carbon dioxide and nitrogen with a Si/Al ratio from 1.5 to 3.5 [7]. Cenospheres in the presence of a large amount of mullite [8,9] show low thermal expansion, considerable thermal stability, and high creep and crack resistance [10]. Their quality depends primarily on the impurities generated in the basic process, i.e., the amount of slag, ash, and unburned mazut or coal [11], while their efficiency depends on cooling rate, melt viscosity, combustion temperature, and falling distance [12,13]. Cenospheres provide an excellent insulating and filling material, used in composite technologies (special rubber mixtures, refractory and insulating materials, ecological barriers, etc.) for the production of light, high strength concrete [14,15,16]. The important features of cenospheres are low bulk density of about 400 kg/m^3^ [17,18], real density of about 2.3 kg/m^3^ [19], low thermal conductivity at room temperature of 0.60 (W/(mK)), and a high melting point of 1200 °C [20,21,22]. The outer diameter of the CS can vary from 1 to 500 µm, with the majority of particles having a size of 60–200 μm, while their total fraction mass in CFA is about 1–2% [23]. It is known from the literature that the water absorption of porous CS is about 18 times greater than that of sand [4,23].

The mineralogical and chemical composition of cenospheres, depending on the concentration of unburned carbon, can change their color from brown to gray or black. Taking into account the ASTM C618 standard, depending on the raw coal, CS are divided into class C from lignite combustion or class F from anthracite combustion. The main role in the activity of CS is played by the combination of silicon, aluminum, and iron oxides, representing 70% of activity for class F and 50% for class C [24,25]. Considering the morphology, elemental composition, and viscosity of slag liquid, researchers have divided cenospheres of A-Si and Fe-Al.-Si alloys into two types: magnetic and non-magnetic [26]. Some CS contain large amounts of Fe exhibiting magnetic properties on the surface, while others are mainly composed of Al and Si [26].

Cenospheres are used in many fields, including building and construction materials [27], plastics [28], ceramics [29], construction [9], coatings [30], lightweight construction materials [31], polymer fillers [32], and energy storage devices [33,34]. Due to their resistance to high temperatures, they are used as a refractory material [35]. In the field of building materials, CS enable a decrease in concrete density while maintaining mechanical strength [36], making them an appropriate production material for lightweight concrete [37]. Additionally, the spherical CS particles act like miniature ball bearings in fresh concrete mix. When added to the mixture in an amount between 1 and 5%, they increase its workability [38].

Souza et al. [39] used CS in the production of lightweight concrete with high tensile strength. Concrete was produced with CS replacing aggregate in quantities of 33, 67 and 100% by volume. It was observed that the addition of CS improved the concrete’s specific strength. Likov et al. [39] found that the crystallization of hydrated products increased with increasing maturation temperature. Hanif et al. [40] developed a formula for a cement composite using aerogel and cenospheres. The obtained material exhibited decreased density. Properties such as particle size and bulk density accelerate the cement hydration process [41]. Satpathy et al. [27] stated that in order to obtain the assumed mechanical properties, it is necessary to determine the optimal percentage of aggregate replacement by cenospheres. Additional thermal and acoustic insulation can be provided by the introduction of spherical CS particles as an additive to concrete, with a positive effect on the insulation properties of plasters, coatings [42], and mortars used in construction [42]. An increase of 100% in the acoustic insulation coefficient was achieved by adding up to 40% CS volume to the cement matrix [43].

A popular use for cenospheres is the production of lightweight structures made of mortar and concrete [38]. The CS particles may be smaller than or similar in size to those of Portland cement and sand, and their shapes may reduce the size and number of open pores. In addition, at higher temperatures (80 °C) CS behave like pozzolana, absorbing free Ca(OH)_2_ from hydrated Portland cement by filling the pores between the cement surface and the CS particles with insoluble silicate hydrates [21,44,45,46]. The pozzolanic activity strengthens the interfacial bonds between the aggregate and the cement matrix. Despite their reactivity in an alkaline environment, they do not exhibit any harmful alkaline–silica reaction effects in mortars [47].

Baronins et al. [48] observed that Portland cement containing cenospheres up to 40% by volume showed a density reduction of 23% compared with cement mortars without CS, and reduced the water absorption capacity to 3%. Mortar with lower water absorption can positively affect the frost resistance and durability of the composite. In addition, it was found that increasing the CS concentration in the range of 0 to 40 percent volume increased the total volume of the open micrometric pores. In cold climates, this in turn may cause cracks in structures to form more quickly during the processes of freezing and thawing.

The main causes of degradation in cement composites at high temperatures are micro-cracks resulting from different properties of the cement matrix and thermal properties of the aggregate [49]. Arizmendi-Morquecho [50] conducted research into the properties of cenosphere-based protective coatings when exposed to high temperatures. The specified thermal expansion coefficient of the microsphere in the temperature range 20–1000 °C reached a value half that of concretes (6.13 × 10^−6^/°C), indicating better thermal compatibility of the cement matrix and the aggregate. The research allowed formulation of a hypothesis about the increased fire resistance of cement composites made with cenospheres. The use of a new lightweight cement composite with low thermal conductivity and high strength may be a promising direction in increasing fire resilience during construction. Cenosphere-based cement composites have been tested at room temperature [51,52,53] and at high temperatures [48,54,55,56,57].

Szymkuć and Tokłowicz [58,59] conducted experimental tests on a light cement composite with the addition of cenospheres. After more than 90 days of maturation, the test samples were heated to temperatures of 105, 200, 400, 600, 700, 800, and 1000 °C. The tests demonstrated a significant improvement in compressive strength compared with traditional concrete. When heated to 60 °C, the samples retained 92% of the initial strength measured at a temperature of about 20 °C, and about 60% when heated to 1000 °C. The obtained material was characterized by low density (1450 kg/m^3^) and reduced thermal conductivity (0.60 W/(mK)) at a temperature of 20 °C. Additionally, numerical analysis was carried out on selected elements of the structure. The results showed an increase in the fire-resistance time for individual parts that were composed of lightweight cement composites that included cenospheres. This indicates the possibility of using lightweight composites in the construction and fire engineering of buildings.

Pursuant to Regulation (EC) No. 1272/2008 (CLP), cenospheres are not classified as hazardous in terms of toxicological and ecotoxic risks [59]. Additionally, they are not classified as hazardous waste according to Commission Decisions 2000/532/EC and 2001/118/EC. If possible, CS should be recovered for further use as part of recycling processes [60,61].

The concept of this research work included evaluating the possibility of using cenospheres as a cement substitute in the production of cement mortars. This research was also aimed at determining the influence of elevated temperatures on designed mortars’ mechanical properties. Annealing temperatures were chosen to correspond to the different phases of fire in rooms of different heights. The selection of temperatures for observation was determined by the points where structural changes of the mortars occurred. The aim of the experiment was to determine the effects of the applied mortar additives on bending strength under compression, under normal and initial thermal load conditions. The heating of the samples was carried out in accordance with the adopted temperature distribution curve, similar to the model curve used in the fire-resistance tests for the structural elements of buildings) [62,63,64].

## 2. Materials and Methods

### 2.1. Cement Mortar with Cenospheres

Cement mortars were made of aggregate with a fraction of 0–2 mm (Kwarcmix, Tomaszów Mazowiecki, Poland), CEM I 42.5 R Portland cement (Cement Ożarów SA, Ożarów, Poland), water, and the addition of cenospheres (Eko Export SA, Bielsko- Biała, Poland).

In order to determine the effect of cenospheres on the mortar strength, two types of samples were prepared: MC—ordinary mortar,MCC—mortar with the addition of cenospheres at 10, 20, and 30% by weight of cement, respectively.

The compositions of cement mortars with different contents of cenospheres are presented in Table 1.

The samples for the experimental tests were made of CEM I 42.5 R cement, meeting the requirements specified in Table 2, and CEN sand. The values provided are the average values for the year 2021 specified by the manufacturer. CEN standard sand is certified compliant with PN EN 196-1: 2016- 07 [64]. It is a natural quartz sand extracted from the “Biała Góra” sand deposits near Tomaszów Mazowiecki. Its grains are rounded, and the mass content of silicon dioxide (SiO_2_) is 98%. The size distribution of standard sand (Table 3) was assumed.

### 2.2. Cenospheres

White cenospheres W-300 with the following chemical composition were used for the tests: Al_2_O_3_ (34–38 wt%), Fe_2_O_3_ (1–3 wt%), SiO_2_ (50–60 wt%), CaO (1–4 wt%), K_2_O (0.1–2 wt%), TiO_2_ (0.5–3 wt%), MgO (0.2–2 wt%). White cenospheres are more resistant to temperature than their gray counterparts. Physical and chemical properties, and the number of individual fractions declared by the producer are presented in Table 4 and Table 5 [66].

### 2.3. Methodology of Testing

The bending strength was determined after 28 days of maturation on 40 × 40 × 160 mm trabeculae, subjected to the bending moment in the three-point bending scheme. During the test, the samples were positioned so that their concreting surface was perpendicular to the direction of the load application. Seven samples of each type were tested.

The mortar consistency was tested by the fall cone method according to PN-EN1015-3: 2000 [68], and density according to PN-85/B-04500. [68]. The consistency determination involved determining the spread of the mortar sample on the shaking table. When assessing the influence of the cenospheres on the properties of the obtained mortars, bending and compressive strength tests were carried out for two maturing periods in accordance with the PN-EN 196-1: 2016-07 standard [69]. Cuboidal samples with dimensions of 40 × 40 × 160 mm were placed horizontally between the breaking supports. They were bent with increasing load until the destruction point. The halves of the beams formed after the test were then used for testing the compressive strength, using the ADVANTEST-9 control and hydraulic console (Controls, Warsaw, Poland).

Water absorption of the mortars was carried out according to PN 85/B-04500 [69], and the heating of samples according to PN-EN 1363-1:2020-07 [70]. The temperature distribution (300 °C, 500 °C, and 700 °C) was similar to conditions corresponding to fire. Thermal distribution in the elements was described by the standard temperature–time curve according to PN-EN 1363-1:2020-07 [68]. 

Annealing (Figure 2) was carried out on 7 samples with dimensions of 40 × 40 × 160 mm, intended for testing the bending strength.

Heating of the hardened cement mortars was carried out at high temperature in accordance with the adopted “temperature–time” distribution. Figure 3 refers to the heating of the samples in a special furnace.

The analysis of the influence of temperature on elements made of the designed cement mortars was carried out over a specified time period, without the cooling phase. After reaching the desired temperatures, i.e., 300 °C, 500 °C, or 700 °C, the samples were further annealed for 30 min to equalize the temperature in the entire volume of the element, followed by free cooling to ambient temperature. The loading time of the elements depended on the temperature (300 °C, 500 °C, or 700 °C) and amounted to 60, 120, or 180 min, respectively. Then, selected mechanical properties of the cement mortars were examined.

### 2.4. Statistical Analysis

Basic statistical analysis was performed using the Statistica 14 analytical tool (TIBCO Software Inc., Palo Alto, CA, USA). Quantitative research was carried out and selected descriptive statistics were presented for the analyzed variables, including measures of occurrence, location, and variability. Using these data, the number of observations, mean results, observation differentiations, etc., were determined.

To predict the compressive and flexural strength of the cenospheres, a regression model was constructed in the form of a unidirectional neural network, described by a non-linear activation function (sigmoidal). Multilayer perceptron networks (MLP networks) consisting of several layers of neurons were trained to determine the neurons’ response signals. The multilayer perceptron solves its task with the use of hyperplanes, by dividing the space of input signals into disjointed areas assigned to different values of the output signals.

## 3. Results

### 3.1. Consistency

It can be stated that cenospheres influence the density and consistency of fresh ordinary mortar. The largest diameter of flow, equal to 14.3 cm, was observed for the ordinary mortar, and the smallest for the mortar 30% C, at 9 cm. The results for 10% C and 20% C mortars were 10.5 cm and 9.5 cm, respectively (Figure 4). On the basis of this research, it was found that the higher the content of cenospheres, the denser was the mortar and the smaller the spreading diameter. The ordinary mortar obtained the highest volumetric density, equal to 2.15 g/cm^3^, the lowest was 1.75 g/cm^3^ in the 30% C mortar. The bulk densities of the 10, 20, and 30% C samples decreased by 7, 12, and 19%, respectively, compared to the standard mortar. Analysis indicated that the increase in the content of microspheres lowered the bulk density of the hardened mortar.

### 3.2. Volumetric Water Absorption 

The highest volumetric water absorption, 20.44%, was achieved by the 10% C mortar, and the lowest, 17.01%, by the ordinary mortar. The mortar with the highest content of cenospheres obtained the lowest result among mortars with additive. The results showed that with the increase in the content of microspheres, the water absorption increased. The opposite was observed in the case of volumetric water absorption—an increase in the proportion of the additive caused a decrease in water absorption. The 30% C mortar achieved 14% higher volumetric water absorption compared with the standard value, and 6% lower than the 10% C mortar. The analysis confirmed that the increase in the proportion of microspheres had a beneficial effect on the volumetric water absorption. Figure 5 shows the results obtained for water absorption.

### 3.3. Compressive and Flexural Strength after 28 and 56 Days of Maturation

The averaged results are shown in Figure 6. The mean compressive strength was calculated from 14 measurements obtained from testing the beam halves. The results are presented in Figure 7. The measurement uncertainty is marked in the graphs in the form of vertical error bars indicating error values of 5%. 

The highest bending strength after 28 days of maturation, equal to 8.4 MPa, was achieved by the ordinary mortar samples, the lowest—2.6 MPa—by the mortar samples in which 30% of the cement was replaced with cenospheres, i.e., MCC30%. Mortars with the lowest content of microspheres (MCC10%) achieved strength 35% lower than the standard average value, while the strength for MCC20% mortars showed a decrease of 48%. The greatest decrease in bending strength, of 69%, was recorded for the MMCC30% samples. For the bending strength after 56 days of maturation, it was observed that the highest strength of 8.0 MPa was achieved by the ordinary standard mortar, and the lowest, equal to 3.3 MPa, by the MCC30% mortar. The composite with the lowest content of cenospheres (MCC10%) obtained strength 32% lower than the ordinary mortar.

When analyzing compressive strength, it was observed in two maturation periods that the strength decreased with the increase in the content of microspheres (Figure 7). The standard mortar obtained the highest strength, equal to 38.8 MPa, while the smallest, 11.3 MPa, was recorded for the MCC30% mortar. Mortar samples with the lowest content of cenospheres (MCC10%) obtained 56% lower strength than the standard samples, while MCC20% mortar showed 60% lower strength compared to MC. After 56 days of maturation, the highest compressive strength of 46.4 MPa was achieved by the ordinary mortar, while the lowest, 14.1 MPa, was achieved by the MCC30% mortar. The best results among composites with microspheres were achieved by mortar samples in which cement was replaced with cenospheres at a quantity of 10%. This series demonstrated 51% lower compressive strength than the ordinary mortar, while the MCC20% samples achieved a 60% lower result. The research showed that the inclusion of cenospheres negatively affected the compressive strength of mortars.

After adding 30% cenospheres, the compressive strength decreased by about 70% after 28 and 56 days of maturation. A similar phenomenon was described in [71]. The decrease in strength was due to the low strength of the cenosphere and the reduction in Portland cement within the mortar. Moreover, according to research [72], the compressive strength of cement-based composites is correlated with porosity. The incorporation of MCC into mortars increases their porosity, because the additive is characterized by a hollow structure. In this study, the bending strength of the mortars gradually decreased when increasing the cenosphere to 30% of the cement mass, despite the fact that MCC has a filling effect and high pozzolanic activity. During the pozzolanic reaction, large quantities of amorphous C-S-H gels and crystal are produced, favoring the formation of hydration products [73,74]. The cenospheric coating in the cement slurry was broken due to the low strength of the MCC during mixing. This phenomenon may explain the decrease in the strength of cement mortars that accompanied the increase in the content of cenospheres. The structure of the transition zone between the cement slurry and the cenospheres was loose, and the density was low. The strength of mortars without cenospheres was higher than that of other cement mortars without addition, at any age, clearly indicating the negative influence of MCC on the mechanical properties of these cement composites [75]. 

### 3.4. Temperature and Material Strength

The average bending strengths are shown in Figure 8. The results were obtained from seven measurements. The average compressive strength was calculated from 14 measurements. The results of the analysis are presented in Figure 9. Error bars with a value of 5% are marked on the chart.

The research showed that the highest bending strength of 3.5 MPa after 28 days of maturing was achieved by mortar samples (MC) heated at 300 °C. Among the mortars with the addition of cenospheres, the MCC20% samples demonstrated the highest bending strength. Compared to the reference samples, the decrease in strength was 25%. Significant amounts of the samples heated at 700 °C were damaged—the mortar fell apart when handled. In terms of bending strength, the strength was found to decrease with increasing temperature. The best strength parameters of 2.4 MPa (MCC10%) and 1.5 MPa (MCC30%) were obtained at 300 °C. Compared with the reference samples, however, these values represented decreases of 31 and 57%, respectively.

After heating at 300 °C, the highest compressive strength, equal to 17.3 MPa, was obtained by the MMC10% mortar, the lowest, equal to 11.8 MPa, by the mortar with the highest amount of additive, i.e., MCC30%. When exposed to a temperature of 500 °C, the results were analogous. The mortar obtained the highest strength, equal to 11.5 MPa, when 10% of cement was replaced with cenospheres. The lowest strength, equal to 5.7 MPa, was obtained by MCC30% mortar. After heating at 700 °C, the highest compressive strength of 5.8 MPa was achieved by the MCC10% mortar, the lowest—2.8 MPa—by the mortar with 30% replacement of cement with additive. It was observed that the highest compressive strength results were achieved by MCC10% mortar, and the lowest by MCC30% mortar. Among the MCC10% mortars, after heating at 300, 500, and 700 °C, compressive strength decreased by 22, 49, and 74%, respectively.

Figure 10 and Figure 11 show the average bending values and compressive strength as a function of temperature. The strengthening effect is indicated for mortars heated to 300 °C with 20 and 30% cenospheres content, amounting to 0.6 for flexural strength and 1.1 for the compressive strength.

### 3.5. Statistical Analysis 

Calculations for two prognostic models were made by the automatic neural networks method (ANN)—one per given strength group. Two input independent variables, i.e., temperature and percentage of cenospheres, and one dependent variable of compressive and bending strength were assumed as predictors, according to the calculations (visible on the last output). The regression pattern was not generated. The five networks with the lowest frequency of errors in learning, testing, and validation were obtained. Table 6 and Table 7 present the calculation results of the ANN models.

The best SANN prognostic models were MLP 2-9-1 (for compressive strength) and MLP 2-8-1 (for flexural strength). The MAPE errors were 10.394% and 13.521% < 15%, respectively. 

The interactive relationships between strength, temperature, and cenospheres content are shown in categorized 3D surface charts. The mapping of the compressive and bending strength as a function of two input variables is presented in Figure 12 and Figure 13. 

## 4. Discussion

In Poland and internationally, research on the use of cenospheres in cement composites has been carried out by only a few research centers, and a very limited number of scientists. As such, limited reliable scientific research is available. In the available literature on the subject, authors suggest the use of fine-grained aggregate with cenospheres, and in the presented research, cement with this additive was mentioned. The conducted research has shown that cenospheric additives affect various parameters of the concrete mix and properties of concrete samples. The current study confirmed that cenospheres reduce the diameter of the flow of the mixture, which was also confirmed by the studies conducted by Baronins et al. [53]. The current research also showed that the addition of cenospheres increased the volumetric absorbability of mortars. This finding was inconsistent with the results of studies published by other authors; in previous works [51,52] it was shown that replacing Portland cement with cenospheres reduced the water absorption capacity to 3%, which may lead to an increase in frost resistance. However, this was not confirmed by the research carried out in this study. The current research also indicates that the addition of cenospheres negatively affects the strength of mortars during compression and bending. As the number of cenospheres increases, strength decreases. This effect was especially noticeable at a temperature of 20 °C. The research to date [36,37,38] has shown that in the production of lightweight concrete with the addition of cenospheres, it is possible to increase both compressive and bending strength, and also high tensile strength. It should also be noted that under the influence of thermal load, Portland cement slurry decomposed the C_4_AH_13_ phase (170–180 °C) and calcium monosulfate (190–200 °C), and the C_3_AH_6_ lost water (340 °C). Ca(OH)_2_ decomposes at 500–550 °C. Further decomposition took place at a temperature of about 750 °C [74,75,76,77].

## 5. Conclusions

This research did not confirm the possibility of the rational use of cenosphere additives in the production of cement mortars. The results and their analysis allow us to formulate the following final conclusions:Cenospheres used as filler at amounts of 10, 20, or 30% of cement mass negatively affected the strength of the samples during bending and shearing. This was noticeable at room temperature of 20 °C and after thermal preloading at 300 °C, 500 °C, and 700 °C.After 28 and 56 days of maturation, the bending strength of mortars with 30% cenospheres was respectively 69% ower than that determined for the reference mortar.The compressive strength determined for samples with 30% addition of cenospheres was lower than that of mortars without addition after 28 and 56 days of maturation, by about 70%.The compressive and flexural strength decreased with increasing annealing temperature and increasing amounts of additive in the mortar in the form of cenospheres.The best SANN prognostic models were MLP 2-9-1 (compressive strength) and MLP 2-8-1 (flexural strength). The estimation showed that the correct regression model was adopted. The MAPE errors were 10.394% and 13.521% < 15%, respectively.

## Figures and Tables

**Figure 1 sensors-22-07518-f001:**
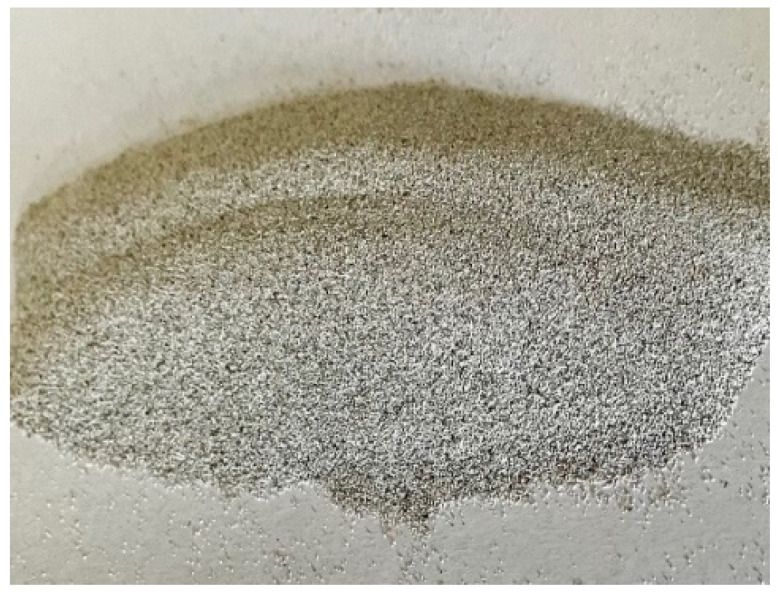
Cenospheres.

**Figure 2 sensors-22-07518-f002:**
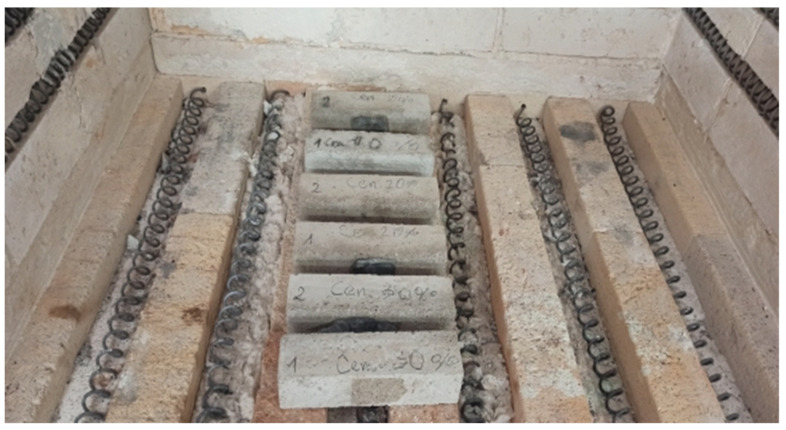
Samples during annealing at 300 °C.

**Figure 3 sensors-22-07518-f003:**
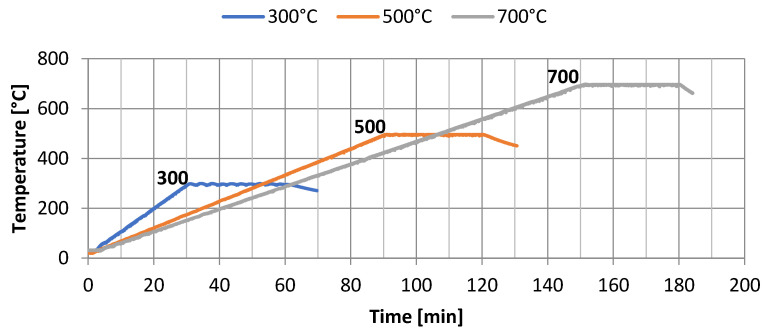
Temperature distribution in the elements.

**Figure 4 sensors-22-07518-f004:**
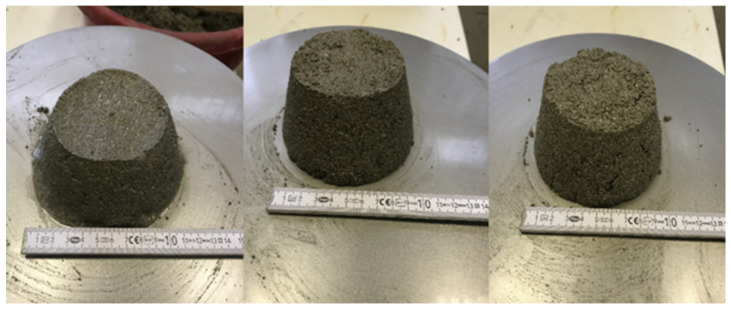
Consistency test—from left to right, 10, 20, and 30% MCC mortars.

**Figure 5 sensors-22-07518-f005:**
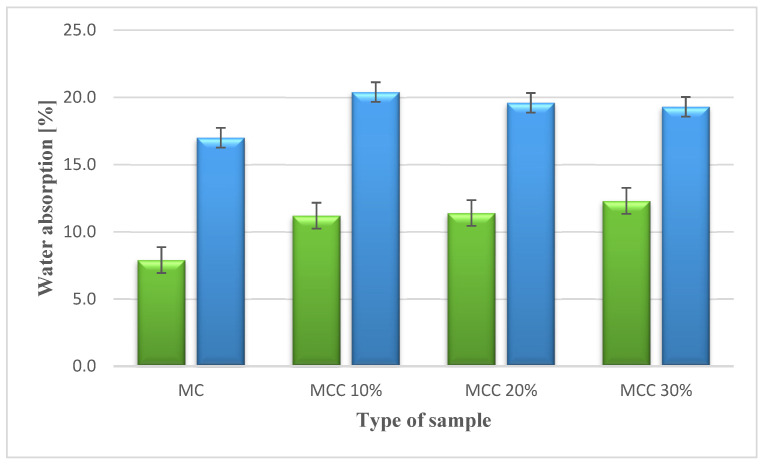
Weight and volume saturation.

**Figure 6 sensors-22-07518-f006:**
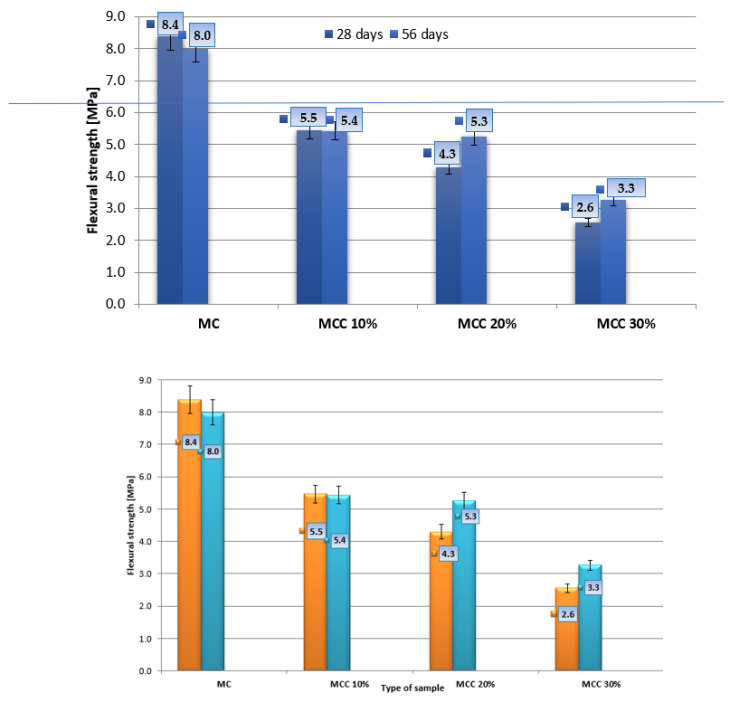
Bending strength after 28 and 56 days of maturation.

**Figure 7 sensors-22-07518-f007:**
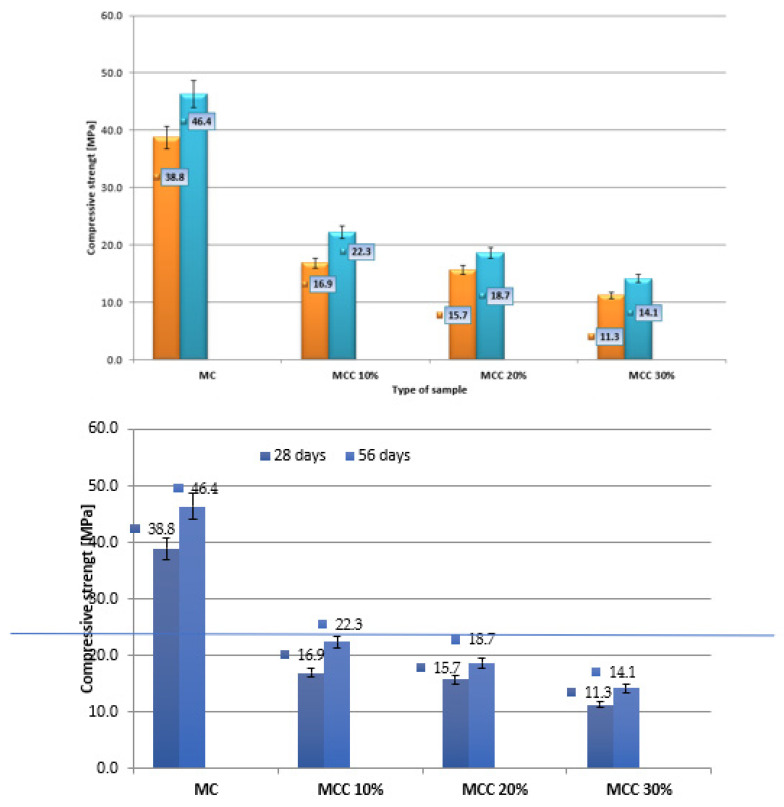
Average compressive strength after 28 and 56 days.

**Figure 8 sensors-22-07518-f008:**
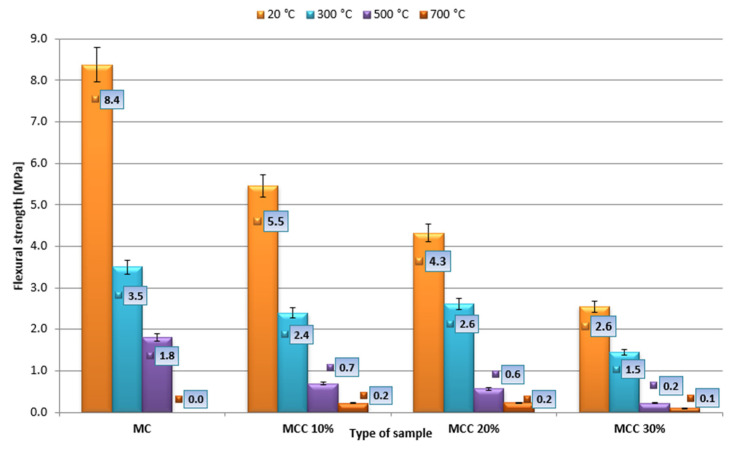
Flexural strength of mortars exposed to heat.

**Figure 9 sensors-22-07518-f009:**
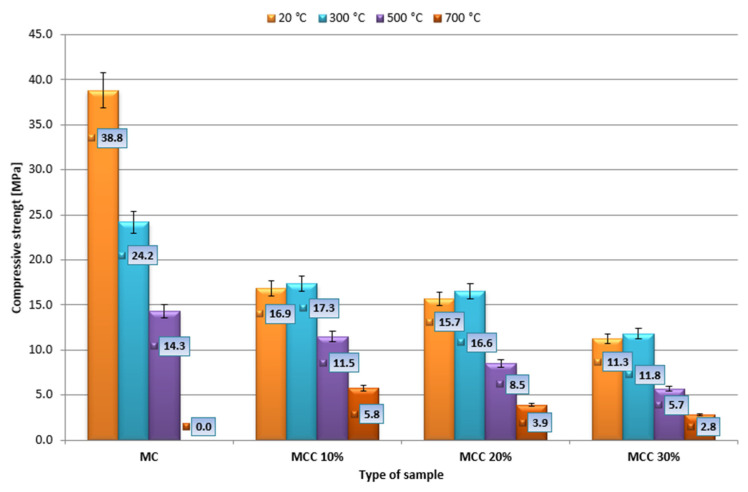
Average compressive strength obtained by the remaining halves of the beams.

**Figure 10 sensors-22-07518-f010:**
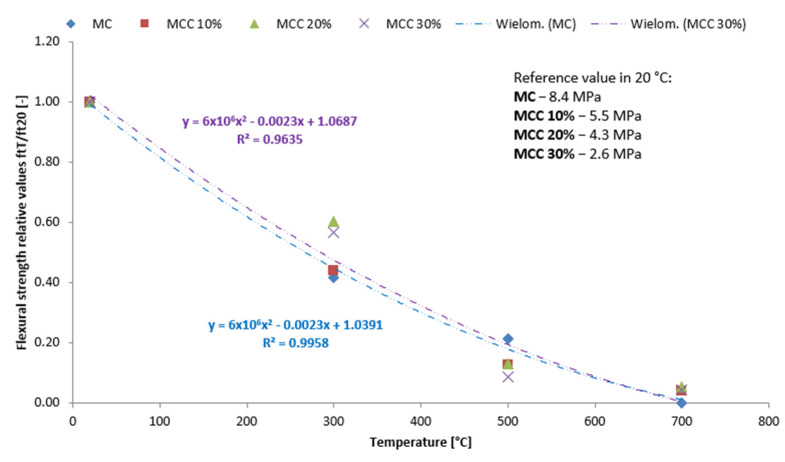
Flexural strength of heated cenospheres: relative ftT/ft20 values [-] with reference ftT values in MPa.

**Figure 11 sensors-22-07518-f011:**
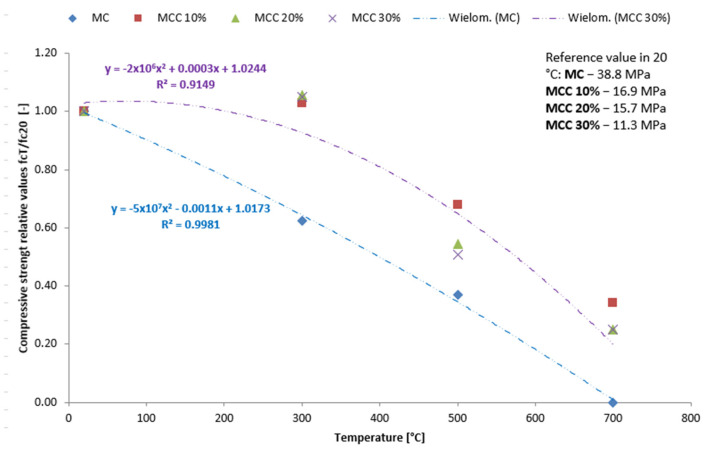
Changes in compressive strength for cenospheres after thermal load: relative fcT/fc20 values [-] with reference fcT values in MPa.

**Figure 12 sensors-22-07518-f012:**
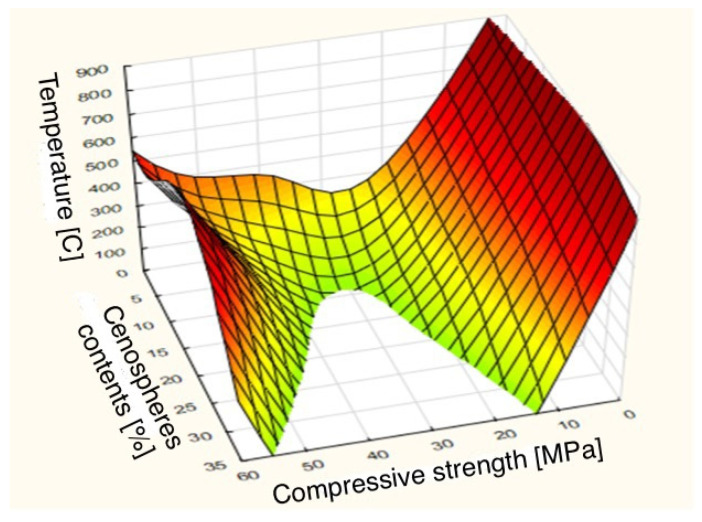
Compressive strength mapping by MLP network.

**Figure 13 sensors-22-07518-f013:**
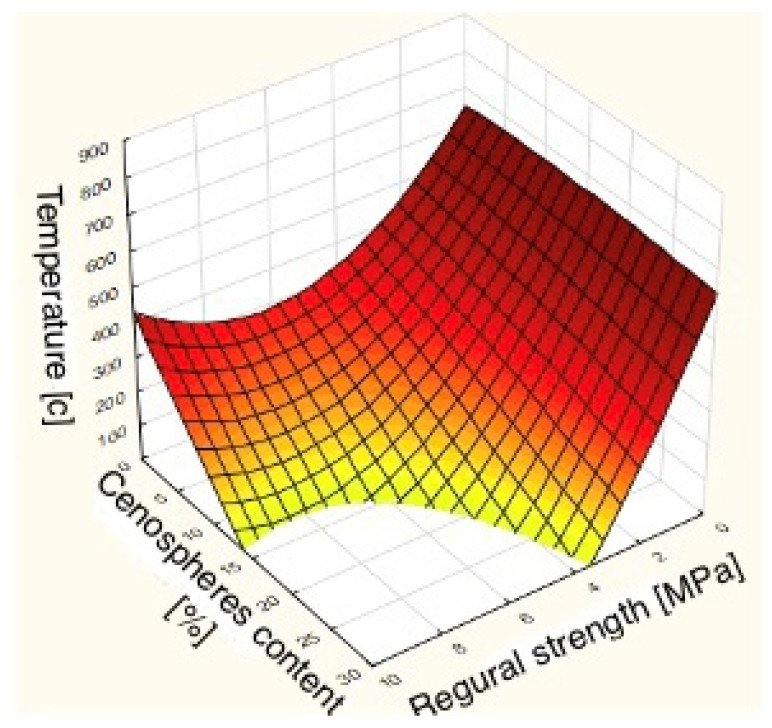
Bending strength mapping by MLP network.

**Table 1 sensors-22-07518-t001:** Cement mortars mix proportions by weight.

Specification	Mass of Concrete Ingredients [g]			
	Aggregate	Water	Cement	Cenosphere
MC	1350.0	225.0	450.0	-
MCC 10%	1350.0	225.0	405.0	45.0
MCC 20%	1350.0	225.0	360.0	90.0
MCC 30%	1350.0	225.0	315.0	135.0

**Table 2 sensors-22-07518-t002:** Chemical, mechanical, and physical properties of cement [65].

Parameters	Standard Requirements PN-EN 197-1	Average Values Achieved
Compressive strength [MPa]		
after 2 days	≥20	29.9
after 28 days	≥42.5, ≤62.5	56.6
Loss on ignition [%]	≤5.0	2.98
The residue is insoluble [%]	≤5.0	0.77
Sulphate content SO_3_ [%]	≤4.0	3.22
Chloride content Cl^-^ [%]	≤0.10	0.05
Start of setting time [min]	≥60	196
Consistency [mm]	≤10	0.5
Specific surface area [cm^2^/g]	no requirements	4138

**Table 3 sensors-22-07518-t003:** Sand granulation distribution [66].

Square mesh side [mm]	2.00	1.60	1.00	0.50	0.16	0.08
Total sieve residue [%]	0	7 ± 5	33 ± 5	67 ± 5	87 ± 5	99 ± 1

**Table 4 sensors-22-07518-t004:** Properties of the cenospheres [67].

Parameters	
Physical condition	Constant
Smell	Almost imperceptible
pH (raw leached cenospheres)	5.5–6.4 (30 min–24 h)
pH (processed leach cenospheres)	6.5–6.9 (30 min–24 h)
Melting temperature	1.260–1.560 °C
Sintering temperature	1300 °C
Optical density	1 g/cm^3^ raw cenosphere
	0.7 g/cm^3^ processed cenospheres (drying at 105 °C)
Water solubility	0.051–0.068 g/L (0.17–0.23% by weight). Remarks: main ingredients do not dissolve in water
Volume density	0.4 g/cm^3^ raw and processed cenospheres (drying at 105 °C)
Bulk density	0.360–0.450 g/cm^3^
Humidity	0.50%

**Table 5 sensors-22-07518-t005:** Particle size specification [67].

	Min%	Max%
+500 μm	0	0
+300 μm	0	4
+150 μm	20	90
+63 μm	5	60
<0.63 μm	0	5

**Table 6 sensors-22-07518-t006:** ANN model for compressive strength.

Index	Net. Name	Training Perf.	Perf Test.	Validation Perf.	Training Error	Test Error	Validation Error	Training Algorithm	Error Function	Hidden Activation	Outupt Activation
1	MLP 2-9-1	0.960985	0.997594	0.981888	10.98046	0.525142	4,710,196	BFGS 35	SAUCE	Exponential	Tanh
2	MLP 2-8-1	0.957127	0.991793	0.982990	12.04051	1.824148	5.194732	BFGS 38	SAUCE	Exponential	Exponential
3	MLP 2-3-1	0.951878	0.993628	0.982332	13,47917	1.532365	5,050,331	BFGS 50	SAUCE	Exponential	Tanh
4	MLP 2-7-1	0.955105	0.994594	0.985742	12.60690	1.231266	3.784681	BFGS 58	SAUCE	Logistic	Identity
5	MLP 2-3-1	0.949582	0.994971	0.992680	14.16327	1.097697	2.615503	BFGS 49	SAUCE	Exponential	Identity

**Table 7 sensors-22-07518-t007:** ANN model for bending strength.

Index	Net. Name	Training Perf.	Perf Test.	Validation Perf.	Training Error	Test Error	Validation Error	Training Algorithm	Error Function	Hidden Activation	Outupt Activation
1	MLP 2-7-1	0.973513	0.989465	0.980619	0.272368	0.126738	0.170937	BFGS 28	SAUCE	Logistic	Logistic
2	MLP 2-7-1	0.978121	0.989089	0.979227	0.221242	0.111589	0.190512	BFGS 30	SAUCE	Logistic	Logistic
3	MLP 2-8-1	0.982814	0.984995	0.981190	0.174354	0.132705	0.171876	BFGS 50	SAUCE	Exponential	Logistic
4	MLP 2-8-1	0.983389	0.985912	0.979608	0.168353	0.125472	0.181284	BFGS 84	SAUCE	Logistic	Exponential
5	MLP 2-3-1	0.982979	0.987203	0.981242	0.173414	0.122459	0.166404	BFGS 53	SAUCE	Exponential	Exponential

## Data Availability

Not applicable.

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
