# Peer review of "Analysis of the Possibility of Using Cenospheres in the Production of Cement Mortars for Use in an Elevated Temperature Environment"

_sensors, 2022, doi:10.3390/s22197518_

Round 1

Reviewer 1 Report

The article reveals the influence of cenospheres on the physical and mechanical properties of mortars. Notes on the article:

1) It is necessary to remove lines 24-32 from the text. The same - 494-504.

2) Table 6 in the article is redundant. This table is the result of a neural network calculation and can be presented as additional material, but not in the article.

3) instead of table. 6 it is better to show the neural network model and the mathematical formulas embedded in this network

4) no explanation why at 700°C exposure, samples MCC 10% - MCC 30% show compressive and flexural strength, in contrast to the control composition

Author Response

Thank You for reviewing our article on the use of cenospheres and for pointing out valuable comments.

1) It is necessary to remove lines 24-32 from the text. The same - 494-504.

In the text, lines 24-32 and 494-504 have been deleted

2) Table 6 in the article is redundant. This table is the result of a neural network calculation and can be presented as additional material, but not in the article.

Yes, we agree with the note, the table shows the neural network.

3) instead of table. 6 it is better to show the neural network model and the mathematical formulas embedded in this network

Table 6 remained removed .

4) no explanation why at 700 ° C exposure, samples MCC 10% - MCC 30% show compressive and flexural strength, in contrast to the control composition

Once again, thank You for the review and we ask for your understanding and favor, because we do not hide, we care very much about the publication

Reviewer 2 Report

The following issues must be corrected, and completed:

Improper use of terminology

Template text remained in chapters Introduction and Materials and methods 

Incorrect use of  English grammar

A significant amount of typos

Very talkative phrasing

Incomplete sentences

Chapters Materials and methods, Results, and Discussion must be developed/evaluated  further

The quality of the graphics/figures must be improved (decimal point, use of appropriate colours, labels of axis)

The reference list is incomplete (check refs 24, 69, 70) 

The citation is sometimes inaccurate, e.g. ref. 82 literature does not mention any transitional or interphase zones.

Detailed remarks are included in the scanned manuscript, please, check it. Research results must be presented in a more precise, straight-to-the-point way. The English of the manuscript needs to be improved as several distractions and inconsistencies remained in it that make reading and interpretation difficult.

Abstract

The following sentences are unnecessary frills, and should be shortened/simplified:

“The topic of research included in this article was to evaluation of the influence..”

“Thanks to the obtained results, the 16 properties of the mortars were compared.”

Line 19: “increasing temperature (20, 300, 500 7000)” – add °C, revise 7000

Line 20: “content of this additive (10, 20, 30%).” specify: wt%

Introduction

Lines 24-38: Guide text of the template, must be deleted.

Line 47: “temperatures (1000-1800oC),” – replace oC with °C

Line 50: “ie plerospheres, ferrospheres and cenospheres - Figure 1.”  - In academic writing, we do not refer to figures in this style. Instead of the hyphen, use an adequate phrase. The correct form of ie is i.e.

Figure 1. Left-sided image: The pictures and letters composing the figure should be enlarged. Function/explanation of the letters should be added to the legend of the figure. Right-sided image: The scale bar is missing.

Line 58: “Gray color is for carbonate, while black color is for inorganic components.”- It should be noted, where does this sentence belong to.

Lines 68, 72: The correct form of ie is i.e.

Lines 69-70: “Cenosphere in the presence of a large amount of mullite [11, 12] shows low thermal expansion, considerable thermal stability, high creep and crack resistance” – Add values/ranges to the listed properties.

Lines 77-78: write “kg/m3” in the correct form: kg/m3

Line 78: “low thermal conductivity at room temperature (W/(mK)” – The value of thermal conductivity is missing.

Line 79: “1200oC” - replace oC with °C

Lines 89-90: “depending on the A-Si and Fe-Al.-Si” – correct A-Si and Fe-Al.-Si

Line 90: “Some CS”- as CS denotes cenosphere, the plural should be written as CSs

Line 91: “high amounts of Fe on surface” – on the surface

Line 95: “polymer fillers [36] , energy” – remove unnecessary space

Line 101: “Souza et al. used” – add the reference number right after the citation

Line 102: “Concrete was produced with a replacement of 33, 67 and 100% by the cenosphere”- The concrete was replaced by the cenosphere? It should be specified, which component of the concrete was replaced. Also the type of % should be specified (wt%, vol%, etc).

Line 103: “the addition of CS improved the concrete specific strength of while maintaining equal mechanical properties” – Revise this sentence. First of all, strength also belongs to the mechanical properties of a material.

Line 104: “Likov et al. they produced” - Likov et al. [] produced (add the reference number right after the citation, delete they)

Line 106: “Hanif et al” - add the reference number right after the citation

Line 109: “Satpathy et al.” - add the reference number right after the citation

Lines 109-111: “in order to achieve the assumed mechanical properties, the optimal percentage of cenospheres utilization should be established, ie 50 and 75%, respectively”- The type of % should be specified (wt%, vol%, etc). The correct form of ie is i.e. Define, respectively to what? Respectively – to what? It is used to relate at least two items previously mentioned in the text. In this case, we have two percentages, but what are the two parameters mentioned previously within the same sentence?

Line 120: “free Ca(OH)2” – use subscription

Line 125: “Sahmenko et al. [53]. Replacing Portland” – write replacing (without capital R)

Line 135: “Arizmendi-Morquecho conducted” - add the reference number right after the citation

Line 138: “range of 20-1000oC … than for concretes (6.13·10-6 1/oC).” - replace oC with °C

Lines 143-144: “increasing resilience fire construction.”-remove line break

Line 146: “Szymkuć, TokÅ‚owicz conducted” - add the reference number right after the citation

Lines 148 and 153: “1000oC” and “of 20oC”- replace oC with °C

Lines 150-151: “After heating to 600°C, the samples retained 92% of their initial strength, and after heating to 1000°C, about 60%.” – Revise this sentence.

Materials and Methods

Lines 175-181: Guide text of the template, must be deleted.

Line 187: “The plan of the experiment of the influence of the cenospheres on…” This sentence contains unnecessary frills, and should be shortened/simplified.

Line 190-191: “mortar with the addition of cenospheres in the amount of 10, 20, 30% of the cement” – Specify, wt% or vol%

Line 196: “Volume composition of” - Volumetric

Line 202: “silicon dioxide (SiO2)” - use subscription

Lines 202-203: “…carried out by sieve analysis on a sample weighing 1350 ± g - Table 3.” Revise this sentence (check 1350  ± g). In academic writing, we do not refer to tables in this style. Instead of the hyphen, use an adequate phrase.

Line 205: “Material properties of cement: chemical, mechanical and physical” – The correct form is: The chemical, mechanical, and physical properties of cement

Table 2.: “Cl-” – Use superscription

Line 210: “with the following chemical composition” - Specify % (wt%, mol%?)

Line 212: “MgO (0.2-2%)- picture 1.” - Do not refer to figures in this style. Instead of the hyphen, use an adequate phrase.

Line 214: “are presented in tables 3, 4” - The correct form is: are presented in Tables 4 and 5”

Line 216: “Physical and chemical properties” – The physical and chemical properties. Complete with the material (properties of the cenosphere)

Table 4: use black letters instead of grey, and capitals for the listed parameters (e.g. Melting, Optical, Water, etc.). The columns of the table are shifted. Remove comma after “1300°C”

Line 218: Write “Particle size specification” instead of sieve analysis, as it is not a result, but the given specification of the material

Lines 221-222: “The consistency of the designed mortars was measured using the drop cone method according to PN-EN1015-3:2000 [71], density according to PN-85/B-04500 [72].” – Revise this sentence.

Line 232: “The seediness of the mortars…” – Define this property. I made a Google search for “mortar seediness”, but no meaningful result was obtained.

Line 235: “exposed to high temperatures, i.e. 20°C, 300°C, 500°C and 700°C” – The correct form is: exposed to the following temperatures: 20°C, 300°C, 500°C, and 700°C. Can 20°C be considered as a high temperature, as you originally wrote? 

Figure 2 is unnecessary

Figure 4. : There are 3 labels (300, 500, and 700°C), but there is only one line.

Line 260: “The analysis and evaluation of the influence of high temperature on the made cement mortars…” - Revise this sentence.

Lines 268-272:  this paragraph belongs to the results.

Lines 268-270: “Under the thermal load, influence the Portland cement slurry decomposes the phase C4 AH13 (170-180°C) and calcium monosulfate (190-200°C), and the C3AH6 loses water (340°C). Ca(OH)2 decomposes at 500-550°C.” - Revise this sentence. Use subscription for the phases.

Results

Lines 290-292: “It is stated that cenospheres influence the individual parameters of fresh cement mortar - density and consistency. The standard mortar had the largest spreading diameter of 291 14.3 cm, the smallest - 9 cm - the 30% C mortar” - Revise these sentences. Delete “however” from the next sentence, as it does not have contradictory information.

Line 291: “The standard mortar..” – In the Materials and methods, it is referred as ordinary mortar. Use the same terms in the manuscript.

Lines 306-307: “The highest volumetric water absorption, equal to 20.44%, was achieved by the mortar 10% C, the lowest - 17.01%, by the standard mortar.” Revise this sentence. Use the same term - standard or ordinary mortar

Lines 313-314: “The study confirmed that the increase in the proportion of microspheres has a positive effect on the volumetric water absorption - Figure 6.” – Use beneficial instead of positive. Do not refer to figures in this style. Instead of the hyphen, use an adequate phrase.

Figure 6: Remove digits from the labels of the y-axis. Use volumetric instead of volume. Use decimal points on the labels of the bars, and more different colours as it is hard to distinguish the bars on the graph. Which data are represented by the squares on the graph? X-axis title is missing.

Lines 319-322: “The bending strength was determined after 28 days of maturation on 40x40x160 mm trabeculae, subjected to the bending moment in the three-point bending scheme. During the test, the samples were positioned in such a way that their concreting surface was perpendicular to the direction of the load application. Seven samples of each type were tested.” – These sentences belong to the test methods. Furthermore, trabeculae are defined as small tissue elements of the body (usually bone or muscle) – are you sure this is the right term for your samples?

Figure 7: Remove digits from the labels of the y-axis. Use decimal points on the labels of the bars, and more different colours as it is hard to distinguish the bars on the graph. Which data are represented by the squares on the graph? In the legend, “28 and 56 dishes of maturation” are mentioned- Are you sure “dish” is the right term? X-axis title is missing.

Line 341: “standard mortar, while the MCC.” – Incomplete sentence, revise it.

Figure 8.: Remove digits from the labels of the y-axis. X-axis title is missing. Use decimal points on the labels of the bars, and more different colours as it is hard to distinguish the bars on the graph. Which data are represented by the squares on the graph? Labels of the y-axis are partially covered by the title.

Lines 346-347: “content of microspheres (Figure 7).” – The related figure is Figure 8.

Lines 350-351: “After 56 maturing dishes, the highest wall strength, equal to 46.4 MPa…” – Revise this sentences (dishes, wall strength).

Line 354: “a 10% exchange” - use replacement instead of exchange

Line 354: “This series showed” – These series

Line 355: “strength result compared” – remove result

Lines 360-361: “and the significant reduction in Portland cement in the mortar” – in the amount of Portland cement

Line 367: “and needle crystal”- crystals

Line 372-373: “In addition, the interphase coming zone is prone to microcracks” – Define this phrase: interphase coming zone. Check the references, as ref [82] does not mention any transition or interphase zone.

Lines 376-377:  “The conducted own research confirms the results presented in the paper.” – Revise this sentence.

Figure 9.: X-axis title is missing. Use decimal points on the labels of the bars, and more different colours as it is hard to distinguish the bars on the graph. Which data are represented by the squares on the graph?

Line 388: delete “own”

Figure 9.: X-axis title is missing. Use decimal points on the labels of the bars, and more different colours as it is hard to distinguish the bars on the graph. Which data are represented by the squares on the graph? “h” is missing from the title of the y-axis (strengt is written)

Line 406: “After baking at” - are you sure this is the right term?

Figure 11.: Use decimal points. Explain/decode ftT, ft20, and wielom.

Figure 12.: Use decimal points. Explain/decode ftT, ft20, and wielom.  Labels of the y- and x-axis are partially covered by the title.

Table 6: What does “ZN” represent? Explain the coding system of the table.

Table 8: Explain/define the terms of the table. Column of Validation error is partially missing.

Lines 448-449: “The MAPE error was obtained = 10.394% and 13.521% <15%, respectively.” – Revise this sentence.

Figures 13 and 14: hard to read the labels

Discussion

The content of this chapter is not a discussion, it must be reworked and rephrased.

Conclusions

The content of this chapter is not suitable, it must be reworked and rephrased.

Lines 474-475: “with increasing temperature (20°C, 300°C, 500°C, 7000°C) and the content of this additive (10, 20, 30%).” - Revise it (7000 °C). Additive content should be specified (wt%, vol%)

Line 483: “The best SANN prognostic model is” – models are

Line 485:  “The MAPE error was obtained = 10.394% and 13.521% <15%, respectively” – Revise this sentence. Define: respectively to what.

Author Response

Thank you for reviewing our article on the use of cenospheres and for pointing out valuable comments.

We have attached the explanations and replies to the comments in the PDF file.

Once again, thank you for the reviews and we ask for your understanding and favor, because we do not hide, we care very much about the publication

Reviewer 3 Report

Dear Sir/Madame,

The submitted manuscript does not represent anything new, while the manuscript itself was made at a very low quality level and as such is not acceptable. The residue after burning coal is slag and fly ash. Both of these waste materials represent aluminosilicate material that has been examined on a significantly higher scientific level on this topic presented by the authors.

Author Response

Thank You for reviewing our article on the use of cenospheres and for pointing out valuable comments.

Both in Poland and in the world, such research on the use of cenospheres in cement composites is carried out by only a few research centers, and therefore a very limited number of scientists. As such, there is a very limited amount of scientific research you can rely on. In the available literature on the subject, the authors mentioned fine-grained aggregate for the cenospheres, and in the presented research, cement with this additive was mentioned. The conducted own research has shown that the addition of cenospheres affects a number of parameters of both the concrete mix and the properties of concrete samples. The studies confirmed that cenospheres reduce the diameter of the mixture flow, which was also confirmed by the research conducted by Baronins , Setin et al. The research also shows that the additions of the cenosphere affect the volume absorbability of mortars by increasing it. This is in contradiction with the results of research published by other authors so far, in the papers it has been shown that the replacement with cenospheres Portland cement reduces the water absorption capacity to 3%, and thus may lead to an increase in frost resistance. However, this was not confirmed in the research carried out in the study. The own research also shows that the addition of cenospheres negatively affects the strength of mortars during both compression and bending. As the number of cenospheres increases, strength decreases. This is especially noticeable at a normal temperature of 20°C. Previous studies have shown that in the production of lightweight concrete with additions of cenospheres make it possible to increase the compressive and bending strength. high tensile strength . It should also be noted that under the influence of thermal load, the Portland cement slurry decomposes the C4 AH13 phase (170-180°C) and calcium monosulfate (190-200°C), and the C3AH6 loses water (340°C). Ca(OH)2 decomposes at 500-550°C. takes place at a temperature of about 750°C [76-79].

Once again, thank You for the review and we ask for your understanding and favor, because we do not hide, we care very much about the publication.

Round 2

Reviewer 1 Report

The authors have corrected the comments of the reviewer

Author Response

Dear Reviewer,

Once again, thank You very much for your kindness and the opportunity to publish the article.

All the Best

Authors

Reviewer 2 Report

Introduction

Lines 69-70: “Cenosphere in the presence of a large amount of mullite [11, 12] shows low thermal expansion, considerable thermal stability, high creep and crack resistance” – Add values/ranges to the listed properties.

Line 78: “low thermal conductivity at room temperature (W/(mK)” – The value of thermal conductivity is missing.

Line 107: guantity - quantity

Line 109: “Likov et al. they produced” - delete they

 Materials and Methods

Lines 202-203: “…carried out by sieve analysis on a sample weighing 1350 ± 5g - Table 3.” In academic writing, we do not refer to tables in this style. Instead of the hyphen, use an adequate phrase e.g. The size distribution of standard sand (Table 3) was carried

Line 224: “with the following chemical composition” - Specify % (wt%, mol%?)

Lines 240-242: “The consistency of the designed mortars was measured using the drop cone method according to PN-EN1015-3:2000 [71], density according to PN-85/B-04500 [72].” – Revise this sentence.

Line 252: “Test water absorption…” – The water absoption

 Results

Line 291: “The standard mortar..” – In the Materials and methods, it is referred as ordinary mortar. Use the same terms in the manuscript.

Lines 315-320: “The standard mortar had the largest spreading diameter of 14.3 cm, the smallest - 9 cm - the 30% C mortar.” “The standard mortar had the highest volumetric density equal to 2.15 g/cm3, the lowest one 1,75g/cm3 to the 30% C mortar.” Revise these sentences.

Lines 448-449: “The MAPE error was obtained = 10.394% and 13.521% <15%, respectively.” – Revise this sentence.

 Conclusions

Line 485:  “The MAPE error was obtained = 10.394% and 13.521% <15%, respectively” – Revise this sentence. Define: respectively to what.

Author Response

Dear Reviewer,

Thank you for the comments that have been taken into account.

Once again, thank You very much for your kindness and the opportunity to publish the article.

All the Best

Authors

Reviewer 3 Report

Dear Sir/Madam,

 In your answer related to my comment as well as according to implemented corrections into last version of the manuscript, you give a reason that I change my opinion into recommended for publication. But according to the your answer, on my comment related to the Fly ash, I wish give just information that on the subject of Fly ash is according to the WoS CoreCollection is already published more than 90,000 papers and of all of them more than 460 published papers are done on the subject of cenosphere. Now when you accept all suggestions as implemented correction into manuscript I can confirm that your arguments, which you give into manuscript, justify publication of it.

 Best Regards

Author Response

(The authors gave the same response as above.)
